# CT Enterography for Preoperative Evaluation of Peritoneal Carcinomatosis Index in Advanced Ovarian Cancer

**DOI:** 10.3390/jcm11030476

**Published:** 2022-01-18

**Authors:** Katty Delgado-Barriga, Carmen Medina, Luis Gomez-Quiles, Santiago F. Marco-Domenech, Javier Escrig, Antoni Llueca

**Affiliations:** 1Department of Radiology, University General Hospital of Castellon, 12004 Castelló de la Plana, Spain; sdomenec@med.uji.es; 2Multidisciplinary Unit of Abdominal Pelvic Oncology Surgery (MUAPOS), University General Hospital of Castellon, 12004 Castelló de la Plana, Spain; luisgomezquiles171261@gmail.com (L.G.-Q.); antonillueca@gmail.com (A.L.); 3Department of Pathology, University General Hospital of Castellon, 12004 Castelló de la Plana, Spain; mcmedina@ono.com; 4Department of General Surgery, University General Hospital of Castellon, 12004 Castelló de la Plana, Spain; 5Department of Medicine, University Jaume I (UJI), 12004 Castelló de la Plana, Spain; javierescrig@telefonica.net; 6Department of Gynecology and Obstetrics, University General Hospital of Castellon, 12004 Castelló de la Plana, Spain

**Keywords:** CT enterography, MDCT, peritoneal carcinomatosis, PCI, ovarian cancer

## Abstract

To compare the diagnostic performance of routine CT (rCT), CT enterography (CTE) and intraoperative quantification of PCI to surgical and pathological reference standards in patients with advanced ovarian cancer, a retrospective study of 122 patients who underwent cytoreduction surgery for ovarian peritoneal carcinomatosis was conducted. Radiological, surgical, and pathological PCIs were obtained from the corresponding reports, and the latter two were considered reference standards. The radiological techniques used were rCT: 64 MDCT (32 × 1 mm) (100 mL iopromide 370 i.v., 800 mL water p.o.), and CTE: 64 MDCT (64 × 0.5 mm) (130 mL iopromide 370 i.v., 1800 mL mannitol solution p.o., 20 mg buscopan i.v.). Data were grouped by imaging technique and analyzed using total PCI and stratified by tumor burden (low-PCI < 10, high-PCI > 20). Agreement, diagnostic performance and degree of cytoreduction were evaluated. Disappointing results for rCT and CTE were obtained when using a surgical referent, but better diagnostic performance and concordance (0.86 vs. 0.78 vs. 0.62, *p* < 0.05) was observed when using a pathological referent—surgical PCI overestimates and overstaged patients. PCI is underestimated by rCT rather than CTE. For high-PCI, the ROC curve was mediocre for CTE and useless for rCT, as it failed to identify any cases. For low-PCI, the ROC was excellent (86% CTE vs. 75% rCT). In four cases with low-PCI as determined by rCT, cytoreduction was suboptimal. CTE has a better diagnostic performance than rCT in quantifying PCI in patients with advanced ovarian cancer, suggesting that CTE should be used as the initial technique. Surgical-PCI could be considered as an imperfect standard reference.

## 1. Introduction

The absence of residual disease after cytoreductive surgery (CRS) is the most important prognostic factor in the management of patients with advanced ovarian cancer (AOC). As a result, the therapeutic approach to these patients is based primarily on disease quantification to decide between neoadjuvant chemotherapy or cytoreductive surgery, so proper patient selection is critical because the complete cytoreductive surgery will depend on this fact.

The peritoneal cancer index (PCI) is a measurement of the volume and extent of peritoneal disease based on a description of the location and size of tumor implants by anatomical region [1]. PCI is an objective and reproducible tool for quantifying tumor burden in AOC patients [2].

The intraoperative PCI estimation is largely subjective; the pathological PCI differs significantly from the surgical PCI and can provide a more precise assessment of the extent of peritoneal cancer; thus, using the pathological PCI will allow for more objective comparisons between different studies [3].

Multidetector computed tomography (MDCT) is the current imaging modality of choice for peritoneal carcinomatosis evaluation. Routine CT (rCT) is the usual technique to assess the extension studies of abdominal neoplasms, its main limitation is the inability to assess mesenteric involvement and small peritoneal deposits in the intestinal serosa [4]. A previous study identified CT enteroclysis (enteral contrast is introduced via a nasojejunal tube placed fluoroscopically prior to CT examination) as a reliable preoperative mapping of the extent and distribution of PC in the small bowel and mesentery [5]. According to another report, using CT enterography (where patients drink oral contrast) as a technical modification would improve the limitations of routine CT [6]. Enteral contrast is employed in both techniques to dilate the intestinal loops. However, enteroclysis requires the placement of a nasojejunal tube under fluoroscopic control, which increases radiation exposure and demands more ward and radiological time.

CT enterography (CTE) is a technique that combines intestinal distension resulting from the administration of a large volume of enteric contrast with the acquisition of sectional images after intravenous contrast administration.

The purpose of this study is to analyze the diagnostic accuracy of routine CT (rCT), CT Enterography (CTE) for the detection of lesions at the regional level (R0 to R8 and R9 to R12) compared with surgical and pathological reference standards, and to compare the agreement and diagnostic performance of routine CT (rCT), CT enterography (CTE) and surgical in PCI scoring with reference standards, in patients with tubo-ovarian peritoneal carcinomatosis.

## 2. Materials and Methods

### 2.1. Participants

This prospective study enrolled 148 consecutive patients treated by cytoreductive surgery in our hospital (a teaching university hospital), who were previously referred to the radiology service for staging or preoperative assessment (routine CT or CT enterography) of peritoneal carcinomatosis, between March 2011 and July 2017. The study was conducted according to the guidelines of the Declaration of Helsinki and approved by the Ethics Committee (approval number: HGCIR03-Feb11) of our hospital and informed consent was obtained from each subject prior to the rCT or CTE examination. Surgical candidates underwent surgery within one or two months of imaging. Tumor characteristics were obtained from the medical records after the pathological examination, including tumor stage, grade, and histologic subtype. Pathological proven tubo-ovarian (epithelial ovarian cancer) or peritoneal cancer (*n* = 128) were inclusion criteria. Six patients were excluded as recurrent ovarian cancer (*n* = 3), the interval between CT and CRS was >30 days (*n* = 1), imaging studies were performed without intravenous contrast (*n* = 1), and there was a lack of surgical or histological reports (*n* = 1). After establishing the selection criteria, we had a sample of 122 consecutive patients. In the first two years of the study, routine CT was performed on all participants (*n* = 42), but following the apparent limitations of this technique, CT enterography was used as an optimized technique from July 2013 (*n* = 80) (Figure 1).

Patient characteristics are summarized in Table 1.

### 2.2. Disease Measurement: Assessment of the Peritoneal Carcinomatosis Index (PCI)

The peritoneal carcinomatosis index (PCI) proposed by Sugarbaker [7] is used to quantitatively evaluate the tumor distribution in the peritoneum. It is based on the quantification of the size of the lesions in 13 abdominopelvic regions. The size of the identified lesions in the peritoneum and visceral serosa of the abdominopelvic organs is converted to nominal quantification values from 0 to 3. LS0 is defined as an undetectable tumor. LS1, LS2, and LS3 describe the maximum diameters of the lesion, 0.5, 5, and >5 cm or confluent, respectively. The sum of the lesion score from all regions determines the PCI value. The highest value will be 39.

### 2.3. Pre-Surgical Radiological Assessment: CT Study Protocols

All CT images were acquired using a 64-MDCT system (Aquilion 64 from Toshiba Medical System). CT examinations were performed in cranio-caudal direction (from the upper thorax aperture to the symphysis pubis) and supine position (Table 2).

#### 2.3.1. Routine CT (rCT)

The patients who went through this technique did not do any previous intestinal preparation. Patients fasted for 4–6 h before imaging. All patients were asked to drink 800 mL of water 30 min before the start of CT examination, on free demand.

Contrast material: After obtaining scout views, 100 mL of IV contrast (iopromide, Ultravist 370^®^, Schering AG, Berlin, Germany) was administered via antecubital vein at 3 mL/s follow by 20 mL of saline solution at same rate by using a power injector (Stellant^®^ D CT, Medrad, Indianola, PA, USA). The datasets were acquired in the portal venous phase (70 s postinjection).

Technique: Routine CT imaging protocol included a 32 × 1 mm collimation, slice thickness of 5 mm with an increment of 5 mm and reconstruction interval of 5 mm; 0.5 s gantry rotation, pitch 1406, 120 kV, automated MA.

No antispasmodic drugs were used.

#### 2.3.2. CT Enterography (CTE)

The patients followed bowel preparation two days before involved both a low-residue diet (first day) and a laxative sodium picosulfate with magnesium citrate formulation (second day).

A neutral enteric contrast agent (mannitol solution 2.5%), was administered orally to achieve small-bowel distention. Outside the scanner room, patient consumed 1800 mL of the oral contrast agent at a steady rate (approximately 300 mL every 10 min) during 1 h. Immediately before scanning, 20 mg of an antiperistaltic agent (scopolaminebutylbromid, Buscapina^®^ 20 mg Boehringer Ingelheim, Ingelheim am Rhein, Germany) was injected intravenously to diminish bowel motion and related artifacts.

Contrast material: A dose of 130 mL of IV contrast (iopromide, Ultravist 370^®^, Bayer) was administered via antecubital vein at a rate of 4 mL/s with a power injector (Stellant^®^ D CT, Medrad); the administration was performed in two phases, in the first phase 100 mL followed by 20 mL of saline solution, and after 55 s, the second phase 30 mL followed by 20 mL of saline solution. The image acquisition started automatically when the attenuation value in the region of interest (ROI) placed in the aorta at the level of the diaphragm reached 100 HU. This protocol allows obtaining a biphasic (portal and arterial phase) single acquisition study.

Technique: CT Enterography imaging protocol included a 64 × 0.5 mm collimation, slice thickness of 3 mm with an increment of 3 mm and reconstruction interval of 3 mm; 0.5 s gantry rotation, pitch 1484, 120 kV, automated MA.

### 2.4. Image Interpretation

The clinical report and PCI score were performed by a single radiologist with experience in abdominal imaging, immediately after the test was performed. The surgical team and the pathologist had access to the clinical report but were blinded to the result of the PCI score.

### 2.5. Intraoperative Assessment

This is determined at the time of the laparotomy. Surgical exploration and palpation are performed to precisely identify the size and distribution of the tumor deposits [8]. It is also checked whether optimal cytoreduction was achieved or not. The PCI assessment was performed by the two experienced surgeons, and the final score was established by consensus.

### 2.6. Pathological Assessment

The pathological anatomy lab quantifies PCI based on the same criteria of tumor size and location; the PCI assessment was performed by the same pathologist. The approach followed a similar pattern to that detailed in another paper [9]. In brief, the surgeon labelled each section of the peritoneum in the en bloc specimen and sent it to the pathologist as a separate specimen. The pathologists discussed the size of the main tumor deposit as well as the presence of other minor deposits in each area. In the case of very small deposits, surgeons marked or sent the suspicious areas separately. One or more sections were extracted from the biggest nodule, as well as one or more sections from confluent deposits. The existence of lymph nodal disease was checked in the subperitoneal fat. Likewise, the omenta were inspected for the existence of lymph nodes. In the absence of gross disease, sections were taken every 2 to 5 cm, but this was not binding.

### 2.7. Data Analysis

To evaluate the diagnostic accuracy of rCT and CTE to detect and localize peritoneal lesions, regional-level analyses for each region was conducted. The presence or absence of disease were compared with both pathological and surgical results. Regions suspected of lesions by CT were rated correct positive if lesion was confirmed by surgical examination and by pathological analysis, respectively; otherwise, it was false positive. Regions not suspected of lesion by CT were rated as correct negative if lesion was not identified by surgical examination and by pathological analysis, respectively; otherwise, it was false negative.

The results of the rCT PCI, CTE PCI and surgical PCI scores were compared to PCI score tabulated at pathology. Patients were categorized as low-PCI (1–10), moderate-PCI (11–20), and high-PCI (>20). These results were compared to the surgical and pathological PCI using the same categories of tumor burden.

### 2.8. Statistical Analysis

#### 2.8.1. Concordance (Agreement)

Lin’s correlation coefficient of agreement (rho_c) measures the agreement on a continuous measurement obtained by two people or methods [10,11]. In this study, it was used to analyze the agreement of the total estimated magnitude of the PCI obtained by each one of the four measurement methods. The concordance correlation coefficient combines precision and accuracy measurements to determine the extent to which observed data deviate from the perfect concordance. The value of the Lin coefficient increases with proximity to the perfect accuracy and precision of the data. Lin coefficient quantifies the agreement ranging from −1 to 1, with perfect agreement at 1. Lin coefficient has the following classification according to strength of agreement (theoretical): >0.99 almost perfect, 0.95–0.99 substantial, 0.90–0.95 moderate, and <0.90 poor [12]. The value of the Lin coefficient increases with proximity to the perfect accuracy and precision of the data. Besides, the Pearson correlation coefficient (r) was calculated as a measure of precision, the bias correction factor
(C_b = rho_c/r)
as the deviation of the measures from perfect accuracy represented by value 1. Kappa statistics was used as a measure of the agreement between evaluators or between methods for the PCI categorized measures.

One way of interpreting the area under the ROC curve is that a test with an area greater than 0.9 has high accuracy, while 0.7–0.9 indicates moderate accuracy, 0.5–0.7, low accuracy and 0.5 a chance result [13].

Pathological and surgical PCIs are considered Gold Standards, though pathological PCI should be considered more accurate between them.

#### 2.8.2. Statistics Summary for a Diagnostic Test

The statistics summary for a diagnostic test (Sensitivity, Specificity, and Area under the ROC curve) were calculated from 2 × 2 tables with each of the stratified PCI measurements against the remaining two. The surgical PCI and the pathological PCI were considered as the best approach to the true disease state. The pathological PCI was considered the most perfect gold standard.

Statistical analysis was performed with Stata Statistical Software Release 15 (College Station, TX, USA: StataCorp LLC).

## 3. Results

### 3.1. Analysis of Diagnostic Performance Based on Lesion Detection at the Regional Level

Considering the regional level analysis, 1586 abdominopelvic regions were tested (rCT: 546 and CTE: 1040). We found a total correlation (in terms of the presence or absence) between rCT and surgical examination in 356 out of 546 regions (64%, 242 out of 378 regions in R0 a R8 and 114 out of 168 regions in R9 a R12, for a total concordance in 86 out of 546 (16%) in a positive sense and in 270 out of 546 (49%) in a negative sense) and a correlation between rCT and pathologic score in 406 out of 546 regions (70%, 264 out of 378 regions in R0 a R8 and 142 out of 168 in R9 a R12, for a total concordance in 80 out of 546 (15%) in a positive sense and in 326 out of 546 (60%) in a negative sense).

Between CTE and surgical examination, we found a total correlation (in terms of the presence or absence) in 786 out of 1040 regions (77.5%, 558 out of 720 regions in R0 a R8 and 228 out of 320 regions in R9 a R12, for a total concordance in 354 out of 1040 (34%) in a positive sense and in 432 out of 1040 (42%) in a negative sense) and a correlation between rCT and pathologic score in 806 out of 1040 regions (77%, 556 out of 720 regions in R0 a R8 and 250 out of 320 in R9 a R12, for a total concordance in 296 out of 1040 (28%) in a positive sense and in 510 out of 1040 (49%) in a negative sense).

As for the diagnostic performance in detecting regional involvement was similar when compared to surgery and pathological anatomy, however CTE performed better than rCT. CTE had a moderate diagnosis accuracy for regions 0 to 8 and the global assessment (regions 0 to 12). CTE had low accuracy in intestinal areas, and rCT could not be estimated given the results obtained (true positives = 0) (Table 3).

### 3.2. Concordance Analysis Based on PCI Determination

Lin’s concordance correlation was <0.90 for all comparisons. Using the cutoffs proposed by McBride [12] this figure indicates poor agreement. The degrees of agreement (rCT and CTE) were lower when using the surgical PCI as the standard reference. The CTE, demonstrated a greater degree of concordance with pathological PCI (rho_c = 0.86) than did surgical and rCT (rho_c = 0.78 y 0.62), although it was still less than the <0.90 cutoff used to define poor agreement (Table 4). According to Pearson’s correlation coefficient, a very strong correlation was observed for all comparisons, regardless of the reference standard used. In terms of categorized tumor burden, a moderate agreement was observed for the CTE-PCI scoring and the surgical-PCI scoring compared to the pathological reference standard.

In general, comparing the results according to the reference standard, we found lower agreement and correlation values when using the surgical analysis.

### 3.3. Staging by PCI Levels: Overestimation, Underestimation, and Agreements

We observed that surgical PCI values overestimated tumor burden, with 35% low-PCI and 44% moderate PCI, while rCT or CTE quantifications underestimated it (Table 5). It was found that the rCT-PCI underestimated the burden in all patients with high-PCI as compared to the surgical and pathological reference standard, while the CTE-PCI correctly diagnosed 23% and 29% (surgical and pathological reference standard) of patients. For any reference standard, rCT and CTE correctly diagnosed 90–100% of patients with low-PCI. In patients with high-PCI, surgical scoring demonstrated a 65% agreement.

### 3.4. Diagnostic Performance Measures in Categorised PCI Scoring

We observed that rCT did not diagnose any patient with high-PCI values, so its diagnostic performance cannot be calculated. For moderate-PCIs, the results were generally poor considering any reference standard (Table 6). Considering surgical PCI values as a reference, both imaging tests showed high sensitivity, but CTE showed better diagnostic performance in patients with low-PCI. The area under the ROC curve indicates moderate accuracy for CTE for identifying low and high-PCI.

Considering pathological PCI values as the gold standard, the surgical PCI scoring technique demonstrated high specificity and sensitivity in the low and high-PCI groups. Imaging tests were more sensitive for low-PCI and more specific for high-PCI. The balance between sensitivity and specificity (AUC) for low-PCI was moderate for all three measurement tests, with the highest accuracy value for CTE followed by surgery, both reaching values close to 90%. Diagnostic accuracy was moderate only for surgical quantification in the high-PCI group, CTE demonstrated low diagnostic accuracy, and rCT could not be calculated.

### 3.5. PCI image Staging and Degree of Cytoreduction

In the 122 cases studied, a high percentage of total cytoreductions was achieved: 90% (110/122) cases. Complete cytoreduction was achieved in 88% (30/34) of patients classified by rCT and 100% (44/44) of those classified by CTE in the low-PCI group. In the high-PCI group, complete cytoreduction was achieved in 75% (6/8) of the patients classified by CTE; however, rCT did not categorize any patients with this tumor burden.

## 4. Discussion

In our study, these two imaging techniques were performed in 122 consecutive patients, showing that CT enterography is more accurate than routine CT in the detection of peritoneal implants in all thirteen regions. In particular, the sensitivity of CTE was significantly higher than that of rCT for the detection of lesions in regions 0 to 8. The specificity of CTE was significantly higher than that of rCT for the detection of peritoneal implants in regions 9 to 12. The major limitation of routine abdominopelvic CT was the inadequate distension of the bowel loops; serous implants and mesenteric nodules were missed or confused with collapsed loops, resulting in the failure to detect lesions in regions 9 to 12. In order to improve this limitation, we used CTE as an optimized technique (Figure 2). CT enterography differs from routine abdominopelvic CT in that it makes use of thin sections and large volumes of enteric contrast material to better display the small bowel lumen and wall [14].

Although CTE is a powerful tool in the evaluation of small bowel disease, our study demonstrates a low diagnostic accuracy in the identification of implants in regions R9 to R12. In relation to the CT technique, Fletcher [15] explains that while initial CT enterography reports used a slice thickness of 5 mm, slice thicknesses between 1 and 3 mm are generally preferred with MDCT as they permit improved contrast and spatial resolution with acceptable tradeoffs and increased image noise. In our study, CTE has proved better diagnostic accuracy than rCT in the detection of implants in the regions (R0 to R8).

In patients with PC, the risk of intestinal subocclusion is higher; however, CTE has been a safe method in these patients. Despite the use of a large volume of oral contrast, only 3% of our patients experienced this problem, which was managed with conservative treatment.

Our study demonstrates a large difference between the results obtained according to the reference standard used, the concordance coefficients and diagnostic performance showed higher values of diagnostic accuracy when compared to the pathological analyses. Previous studies [3,16] have concluded that the surgical PCI is not quite objective and should be considered an imperfect standard of reference for evaluating the performance of another test. The surgical PCI has demonstrated a clear tendency to overestimate tumor burden and, as a result, overstates patients. Despite this, compared to radiological tests, it showed the best diagnostic performance, both for identifying patients with a low and high tumor burden.

The PCI estimation of both radiological techniques showed a strong correlation with surgical and pathological quantification, allowing for reliable preoperative tumor burden prediction. Our results are in accordance with previous research which indicated that radiological studies underestimate the PCI score; rCT underestimated the score in most patients with moderate-PCI and in all those with high-PCI; however, poor agreement was demonstrated for Lin’s concordance correlation and moderate for Kappa agreement. CTE has showed the highest concordance values, followed by surgery.

The prognostic impact of tumor burden categorization based on PCI < 10 and PCI > 20 has been described as an independent prognostic factor, with improved survival rates in patients with low-PCI [2]. CTE demonstrated better diagnostic performance than rCT in the identification of patients with low-PCI. In cases of high-PCI, CTE has demonstrated low accuracy but was superior to rCT.

Achieving complete cytoreduction is the most important independent risk factor for predicting survival [17]. In our results, optimal cytoreduction was achieved in all low-PCI cases classified by CTE, in contrast to those classified by rCT, which had a small percentage of cases with suboptimal cytoreduction.

The importance of quantification and screening tools is to identify patients who can undergo complete cytoreduction, PCI > 20 is a predictor of incomplete surgical resectability [1,18,19]. Only one patient had suboptimal cytoreduction in patients with high-PCI detected by CTE; we must consider that PCI is a reliable tool, but it is not the only one useful in selection, so future studies should evaluate the diagnostic efficiency of CTE to identify other criteria for unresectability. This study shows that rCT is incapable of recognizing high-PCI and, as a result, cannot predict the success of cytoreduction.

Recent studies demonstrate the utility of functional imaging techniques, such as PET-CT and diffusion MRI, in the identification of peritoneal lesions and as a method for selecting candidates for cytoreductive surgery [20]. The CT scan has a clear limitation over functional imaging as a morphological imaging technique. Since CT is the most widely used modality in the assessment of peritoneal image dissemination due to its broad availability and speed of acquisition [21], we can investigate and determine when the use of functional tests is appropriate. A previous meta-analysis [22] emphasized the importance of using a specific CT protocol to detect PC, our study supports the use of CTE in the initial assessment of patients with AOC.

Our study had several limitations. First, our study design introduced some confounding factors; the images were read by a single radiologist. Previous studies [6,23] demonstrated the impact of experience in image interpretation in patients with peritoneal carcinomatosis; the radiologist’s experience was zero in rCT cases, as opposed to two and a half years of experience in CTE cases. The slice thickness was not the same in the two radiological tests. Only in the CTE group were antiperistaltic agents used to eliminate peristalsis and reduce motion artefacts. Another limitation was that the patients had the same tumor origin but could have different tumor extensions between the two groups in the experiment.

## 5. Conclusions

CT enterography appears to be more accurate than rCT in detecting peritoneal lesions in all regions and in scoring PCI especially in patients with low tumor burden. This study supports the use of CTE rather than rCT as the initial technique for assessing patients with tubo-ovarian peritoneal carcinomatosis and suggests that the surgical PCI score is an unsatisfactory scale for evaluating new imaging tests.

## Figures and Tables

**Figure 1 jcm-11-00476-f001:**
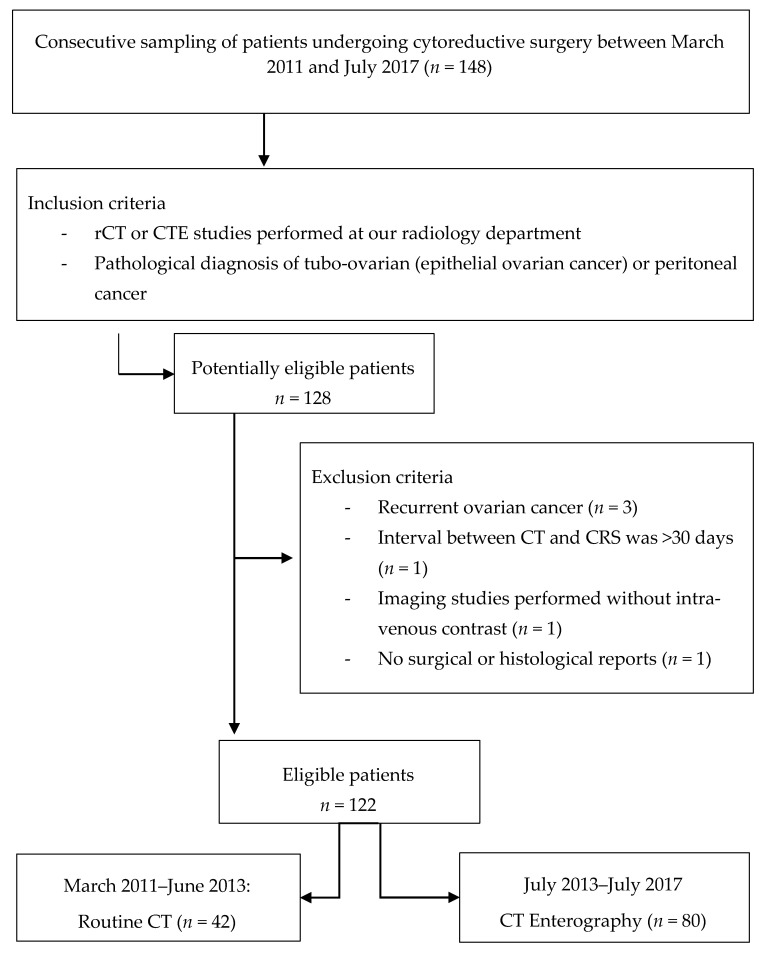
Flow diagram of the participants included in the study.

**Figure 2 jcm-11-00476-f002:**
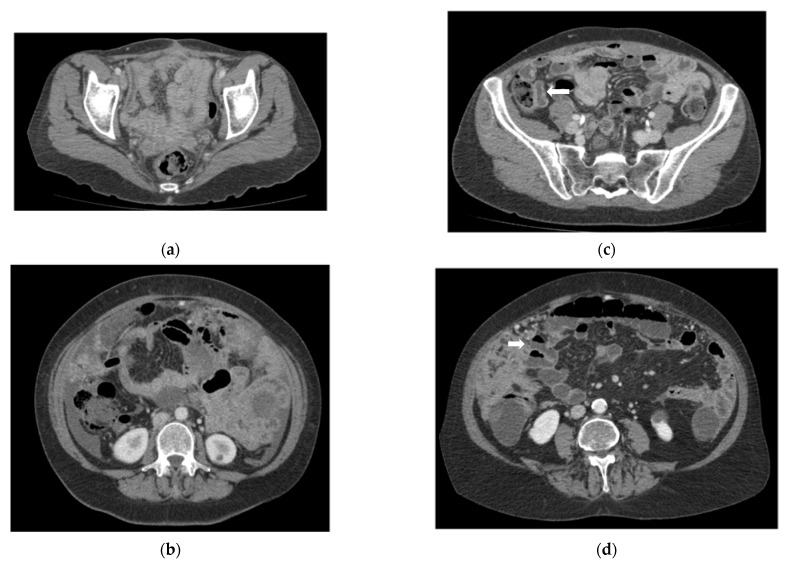
Axial slice images from different patients are shown to compare the degree of intestinal distention obtained by: (**a**,**b**) routine CT; (**c**,**d**) CT enterography section at the pelvis and mid abdomen of a patients with peritoneal carcinomatosis show the difference between the non-optimized and optimized imaging techniques, with the CTE an adequate distention of the intestinal loops is achieved. (rCT: routine CT, CTE: CT enterography), soft tissue nodules (arrowhead).

**Table 1 jcm-11-00476-t001:** Clinicopathologic and general characteristics of the 122 patients with advanced ovarian cancer treated by cytoreductive surgery.

	rCT (*n* = 42)	CTE (*n* = 80)
Age (years)		
Median	60	65
Range	27–44	31–84
CA 125		
Median	276	368
Range	27–890	30–1200
Tumor origin		
Ovarian	40 (95.2%)	76 (95%)
Fallopian tube	2 (4.8%)	2 (2.5%)
Peritoneum	0 (0%)	2 (2.5%)
Histologic findings		
Serous carcinoma	32 (76.2%)	64 (80%)
Mucinosus carcinoma	4 (9.5%)	6 (7.5%)
Endometrioid carcinoma	6 (14.3%)	10 (12.5%)
Adverse Effects on CT		
None	-	106 (87%)
Nausea	-	9 (7%)
Diarrhea	-	4 (3%)
Intestinal subocclusion	-	4 (3%)
Radiological PCI		
1–10	34 (81%)	44 (55%)
11–20	8 (19%)	28 (35%)
>20	0 (0%)	8 (10%)
Surgical PCI		
1–10	20 (47.6%)	26 (32.5%)
11–20	14 (33.4%)	28 (35%)
>20	8 (19%)	26 (32.5%)
Pathological PCI		
1–10	30 (71.5%)	38 (47.5%)
11–20	8 (19%)	28 (35%)
>20	4 (9.5%)	14 (17.5%)
Cytoreduction		
CC-0	34 (81%)	76 (95%)
CC-1 (<2 cm)	4 (9.4%)	4 (5%)
CC-2 (2.6–5 cm)	2 (4.8%)	0 (0%)
CC-3 (>5 cm)	2 (4.8%)	0 (0%)

**Table 2 jcm-11-00476-t002:** Characteristics of the CT study protocols.

Routine CT (rCT)		CT Enterography (CTE)
Aquilion 64 Toshiba	Equipment	Aquilion 64 Toshiba
32 × 1 mm	Collimation	64 × 0.5 mm
5 mm	Slice thickness	3 mm
5 mm	Reconstruction interval	3 mm
100 mL	Intravenous (IV) contrast	130 mL
Portal phase		Biphasic (one-time acquisition)
No	Intestinal preparation	Low-residue diet + laxative formulation
800 mL	Oral contrast quantity	1800 mL
Water	Oral contrast	Solution Mannitol 2.5%
Free demand	Frequency of administration (oral contrast)	300 mL every 10–20 min
No	Spasmolytic	Buscapina^®^

**Table 3 jcm-11-00476-t003:** Comparative diagnostic performance between rCT and CTE vs. surgical and pathological examination at regional level analyses.

		Sensitivity (95% CI)		Specificity (95% CI)		AUC (95% CI)	
		Surg	Path	Surg	Path	Surg	Path
R0 a R12	rCT	32% (24–40)	39% (29–48)	97% (94–100)	96% (93–99)	65% (59–70)	67% (61–74)
	CTE	64% (58–70)	71% (65–77)	89% (85–93)	82% (77–86)	76% (71–81)	76% (71–82)
R0 a R8	rCT	40% (31–49)	44% (34–55)	95% 90–100)	93% (88–98)	68% (61–75)	69% (61–76)
	CTE	73% (67–79)	77% (70–83)	84% (78–90)	78% (72–84)	78% (72–84)	77% (71–83)
R9 a R12	rCT	NC *	NC *	NC *	NC *	NC *	NC *
	CTE	39% (28–51)	44% (28–61)	97% (93–100)	88% (82–94)	68% (60–75)	66% (55–77)

* NC: Not calculable: No regional lesions detected. 95% CI: 95% confidence interval (null value: 50%). R0: Region 0. R8: Region 8, R9: Region 9, R12: Region 12. PCI: Peritoneal Cancer Index, rCT: routine CT, CTE: CT Enterography, Surg: Surgical scoring, Path: Pathological scoring.

**Table 4 jcm-11-00476-t004:** Agreement and correlation analysis between PCI quantifications, routine CT or CT enterography, using surgical and pathological PCI values as reference standards.

	Surgical PCI		Pathological PCI		
	rCT	CTE	rCT	CTE	Surgical PCI
Lin (rho_c)	0.49	0.65	0.62	0.86	0.78
Lin *p* value	*<0.001*	*<0.001*	*<0.001*	*<0.001*	*<0.001*
Pearson (r)	0.83	0.77	0.80	0.87	0.85
Pearson *p* value	*<0.001*	*<0.001*	*0.002*	*<0.001*	*<0.001*
C_b	0.59	0.84	0.78	0.98	0.82
Total agreement *	0.52	0.48	0.71	0.70	0.64
Kappa (agreement not due to chance) *	0.14	0.21	0.26	0.50	0.44
Kappa *p* value *	*0.160*	*0.020*	*0.070*	*<0.001*	*<0.001*

* For categorized PCI 1–10; 11–20; >20. Italics are used to display *p* value.

**Table 5 jcm-11-00476-t005:** Staging by tumor burden (Low-PCI, Mod-PCI, High-PCI). Concordance (≈), underestimation (⇓), overestimation (⇑) using two gold standards: surgical and pathological scoring.

		Low-PCI		Mod-PCI		High-PCI	
		Surg	Path	Surg	Path	Surg	Path
Low-PCI 1–10	rCT	(≈) 100%	(≈) 93%	(⇓) 86%	(⇓) 75%	(⇓) 25%	-
		20	28	12	6	2	0
	CTE	(≈) 92%	(≈) 95%	(⇓) 64%	(⇓) 29%	(⇓) 8%	-
		24	36	18	8	2	0
	Surgical		(≈) 65%	-	(⇓) 6%	-	-
			44	0	2		0
Mod-PCI 11–20	rCT	-	(⇑) 7%	(≈) 14%	(≈) 25%	(⇓) 75%	(⇓) 100%
		0	2	2	2	6	4
	CTE	(⇑) 3%	(⇑) 5%	(≈) 29%	(≈) 57%	(⇓) 69%	(⇓) 71%
		2	2	8	16	18	10
	Surgical		(⇑) 32%	-	(≈) 50%		(⇓) 11%
			22	0	0		2
High-PCI >20	rCT	-	-	-	-	-	-
		0	0	0	0	0	0
	CTE	-	-	(⇑) 7%	(⇑) 14%	(≈) 23%	(≈) 29%
		0	0	2	4	6	4
	Surgical		(⇑) 3%		(⇑) 44%		(≈) 89%
			2		16		16

The table shows the correlation between the PCI measurements tools (rows): rCT (*n* = 42), CTE (*n* = 80), Surg (*n* = 122) and the surgical and pathological reference standards (columns), with demonstration of number of agreements, overestimate, and underestimate cases. PCI: peritoneal cancer index, rCT: routine CT, CTE: CT enterography, Surg: surgical scoring, Path: pathological scoring.

**Table 6 jcm-11-00476-t006:** Comparative diagnostic performance between the PCI measurements tools according to the tumor burden (Low-PCI, Mod-PCI, High-PCI) and compared with surgical and pathological reference standard.

		Sensitivity (95% CI)		Specificity (95% CI)		AUC (95% CI)	
		Surg	Path	Surg	Path	Surg	Path
Low-PCI 1–10	rCT	100% (69–100)	100% (78–100)	27% (6–61)	50% (12–88)	64% (50–77)	75% (53–97)
	CTE	92% (64–100)	94% (73–100)	63% (42–81)	77% (55–92)	78% (66–90)	86% (75–96)
	Surg	-	68% (48–82)	-	96% (82–100)	-	82% (73–90)
Mod-PCI 11–20	rCT	25% (63–81)	25% (63–81)	65% (38–86)	82% (57–96)	45% (18–72)	54% (28–80)
	CTE	29% (8–58)	57% (29–82)	62% (41–80)	77% (56–91)	45% (30–61)	67% (51–83)
	Surg	-	50% (26–74)	-	72% (56–85)	-	61% (47–75)
High-PCI >20	rCT	NC *	NC *	NC *	NC *	NC *	NC *
	CTE	75% (19–99)	50% (7–93)	72% (55–86)	76% (81–95)	74% (48–99)	68% (39–97)
	Surg	-	89% (52–100)	-	83% (70–92)	-	86% (74–98)

* NC: Not calculable: No PCI > 20 cases diagnosed by rCT; 95% CI: 95% confidence interval (null value: 50%). The pathological PCI scoring was considered the most perfect gold standard. PCI: peritoneal cancer index, rCT: routine CT, CTE: CT enterography, Surg: surgical scoring, Path: pathological scoring.

## Data Availability

The data that support the findings of this study are available from the corresponding author upon reasonable request.

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
