# Peer review of "CT Enterography for Preoperative Evaluation of Peritoneal Carcinomatosis Index in Advanced Ovarian Cancer"

_jcm, 2022, doi:10.3390/jcm11030476_

Round 1

Reviewer 1 Report

Thank you for this opportunity to review this interesting study. Preoperative assessment of the extent of PM can be challenging, the authors appropriately address this in a study that shows improved performance of CT enterography for determining the extent of PM as compared to routine CT in a before and after set-up. They also show an interesting secondary outcome which suggests that the extent of PM as determined at surgical exploration may be imperfect and overestimates the extent of PM. I do have some comments which I will elaborate here below:

  • The AUCs from ROC analysis as well as diagnostic parameters (like sens/spec) are not percentages or ratios but a summary statistic ranging from 0-1, please also present with 95% confidence intervals.
  • Although Lin’s concordance correlation coefficient may not be wrong in this situation, I suggest using the more common intraclass coefficient (ICC) which is very similar but would allow for a better comparison with other literature. Please also consider showing 95% confidence intervals. 
  • You describe in the paper that the study was approved by medical ethics review and informed consent was obtained from the patient. But I am unclear as to what the patient consented to? Was the switch to the routine use of CTE anticipated in the study protocol? Or was the study set up with another endpoint in mind? Please elaborate.
  • Please provide the experience and expertise of the sole reporting radiologist. Consider including a second blinded retrospective reader. 
  • The benefit of CTE could be attributed to various differences in the imaging and preparation protocols, consider elaborating which could have contributed the most. 
  • Did any patients receive neoadjuvant chemotherapy?
  • Why classify the PCI into three categories? Trends regarding over and underestimation between two tests can better be assessed using a bland-altman plot.

Author Response

Response to Reviewer 1 Comments

General comments:

Thank you for this opportunity to review this interesting study. Preoperative assessment of the extent of PM can be challenging, the authors appropriately address this in a study that shows improved performance of CT enterography for determining the extent of PM as compared to routine CT in a before and after set-up. They also show an interesting secondary outcome which suggests that the extent of PM as determined at surgical exploration may be imperfect and overestimates the extent of PM. I do have some comments which I will elaborate here below:

The AUCs from ROC analysis as well as diagnostic parameters (like sens/spec) are not percentages or ratios but a summary statistic ranging from 0-1, please also present with 95% confidence intervals.

The diagnostic parameters used are percentages, and the confidence intervals are included in the respective tables.

Table 6. Comparative Diagnostic Performance Between the PCI Measurements Tools According to the Tumor Burden (Low-PCI, Mod-PCI, High-PCI) and Compared with Surgical and Pathological Reference Standard.

Sensitivity

(95% CI)

Specificity

(95% CI)

AUC

(95% CI)

Surg

Path

Surg

Path

Surg

Path

Low-PCI

1–10

rCT

100%

(69 - 100)

100%

(78 - 100)

27%

(6 - 61)

50%

(12 - 88)

64%

(50 - 77)

75%

(53 - 97)

CTE

92%

(64 - 100)

94%

(73 - 100)

63%

(42 - 81)

77%

(55 -92)

78%

(66 - 90)

86%

(75 - 96)

Surg

-

68%

(48 - 82)

-

96%

(82 - 100)

-

82%

(73 - 90)

Mod-PCI

11-20

rCT

25%

(63 - 81)

25%

(63 - 81)

65%

(38 - 86)

82%

(57 - 96)

45%

(18 - 72)

54%

(28 - 80)

CTE

29%

(8- 58)

57%

(29 - 82)

62%

(41 - 80)

77%

(56 -91)

45%

(30 - 61)

67%

(51 - 83)

Surg

-

50%

(26 - 74)

-

72%

(56 - 85)

-

61%

(47 - 75)

High-PCI

>20

rCT

NC*

NC*

NC*

NC*

NC*

NC*

CTE

75%

(19 - 99)

50%

(7 - 93)

72%

(55 - 86)

76%

(81 - 95)

74%

(48 - 99)

68%

(39 - 97)

Surg

-

89%

(52 - 100)

-

83%

(70 - 92)

-

86%

(74 - 98)

*NC: Not calculable: No PCI>20 cases diagnosed by rCT. 95% CI: 95% confidence interval (null value: 50%). The pathological PCI scoring was considered the most perfect gold standard. PCI: Peritoneal Cancer Index, rCT: routine CT, CTE: CT Enterography, Surg: Surgical scoring, Path: Pathological scoring.

Although Lin’s concordance correlation coefficient may not be wrong in this situation, I suggest using the more common intraclass coefficient (ICC) which is very similar but would allow for a better comparison with other literature. Please also consider showing 95% confidence intervals. 

The idea of using the intraclass coefficient is an excellent contribution. However, we would have to reformulate the whole statistical work and that exceeds the review time. Table 1 describes the different correlation coefficients we have used. 

Table 4. Agreement and Correlation Analysis Between PCI Quantifications, Routine CT or CT Enterography, Using Surgical and Pathological PCI Values as Reference Standards.

Surgical PCI

Pathological PCI

rCT

CTE

rCT

CTE

Surgical PCI

Lin (rho_c)

0.49

0.65

0.62

0.86

0.78

Lin valor p

< 0.001

< 0.001

< 0.001

< 0.001

< 0.001

Pearson (r)

0.83

0.77

0.80

0.87

0.85

Pearson valor p

< 0.001

< 0.001

0.002

< 0.001

< 0.001

C_b

0.59

0.84

0.78

0.98

0.82

Total agreement*

0.52

0.48

0.71

0.70

0.64

Kappa (agreement not due to chance) *

0.14

0.21

0.26

0.50

0.44

Kappa value p*

0.160

0.020

0.070

<0.001

<0.001

* For categorised PCI 1-10; 11-20; >20.

You describe in the paper that the study was approved by medical ethics review and informed consent was obtained from the patient. But I am unclear as to what the patient consented to? Was the switch to the routine use of CTE anticipated in the study protocol? Or was the study set up with another endpoint in mind? Please elaborate.

In our centre, prospective studies require patient approval for data processing.

Please provide the experience and expertise of the sole reporting radiologist. Consider including a second blinded retrospective reader. 

The radiologist's experience was zero in rCT cases, as opposed to two and a half years of experience in CTE cases. We will consider a second retrospective reader for further work. Thank you for your input.

The benefit of CTE could be attributed to various differences in the imaging and preparation protocols, consider elaborating which could have contributed the most. 

Adequate distension of bowel loops and thinner slices contribute to better technical performance.

Reviewer 2 Report

The manuscript entitled, “CT Enterography for Preoperative Evaluation of Peritoneal Carcinomatosis Index in Advanced Ovarian Cancer” is not worth acceptance now.

I concerned about the limitation of this study. The author described that the slice of thickness was different in routine CT and CT enterography. This is directly connected to the result of this study so that it cannot be acceptable. If the condition of CT scan was adjusted to the routine CT and the results were re-investigated, the manuscript would be worth review.

Regional level (R0 to R12) were not define in the study.

In Table 1, Histologic findings such as serous cystoadenocarcinoma etc, were no more used in WHO classification. They should be corrected to the latest one.

Author Response

Response to Reviewer 2 Comments

General comments:

The manuscript entitled, “CT Enterography for Preoperative Evaluation of Peritoneal Carcinomatosis Index in Advanced Ovarian Cancer” is not worth acceptance now.

I concerned about the limitation of this study. The author described that the slice of thickness was different in routine CT and CT enterography. This is directly connected to the result of this study so that it cannot be acceptable. If the condition of CT scan was adjusted to the routine CT and the results were re-investigated, the manuscript would be worth review.

Regional level (R0 to R12) were not define in the study.

In Table 1, Histologic findings such as serous cystoadenocarcinoma etc, were no more used in WHO classification. They should be corrected to the latest one.

Response:

Thank you for taking the time to review our work. I fully agree with your observation about the differences between the slice thickness of the two tests evaluated; both the slice thickness and the use of neutral oral contrast influence the results; both combined constitute CT Enterography. It is evident that a thinner slice thickness test has better results. However, CT enterography is not the technique of choice in the initial assessment of patients with peritoneal carcinomatosis. 

In the present study, we did not define the regions used for the calculation of PCI as they are widely known and we did not want to overdo the content; if you consider the publication, we could define them.

rCT (n=42)

CTE (n=80)

Age (years)

Median

60

65

Range

27 – 44

31 – 84

CA 125

Median

276

368

Range

27 – 890

30 - 1200

Tumor origin

Ovarian

40 (95,2%)

76 (95%)

Fallopian tube

2 (4,8%)

2 (2,5%)

Peritoneum

0 (0%)

2 (2,5%)

Histologic findings

Serous carcinoma

32 (76,2%)

64 (80%)

Mucinosus carcinoma

4 (9,5%)

6 (7,5%)

Endometrioid carcinoma

6 (14,3%)

10 (12,5%)

Adverse Effects on CT

None

-

106 (87%)

Nausea

-

9 (7%)

Diarrhea

-

4 (3%)

Intestinal subocclusion

-

4 (3%)

Radiological PCI

1-10

34 (81%)

44 (55%)

11-20

8 (19%)

28 (35%)

>20

0 (0%)

8 (10%)

Surgical PCI

1-10

20 (47,6%)

26 (32,5%)

11-20

14 (33,4%)

28 (35%)

>20

8 (19%)

26 (32,5%)

Pathological PCI

1-10

30 (71,5%)

38 (47,5%)

11-20

8 (19%)

28 (35%)

>20

4 (9,5%)

14 (17,5%)

Cytoreduction

CC-0

34 (81%)

76 (95%)

CC-1 (<2cm)

4 (9,4%)

4 (5%)

CC-2 (2,6-5 cm)

2 (4,8%)

0 (0%)

CC-3 (>5cm)

2 (4,8%)

0 (0%)

Reviewer 3 Report

  Dear Authors, I believe that this paper has tried to compare different types of accuracy indices in relation to the preoperative evaluation of patients with peritoneal carcinomatosis from advanced ovarian cancer. This is an elegant paper that demonstrates the potential and limits of CT-mediated Enterography. I suggest the authors to correct some typos.

Author Response

Thank you for taking the time to review our work. We can make corrections where appropriate, please attach them to the text.

Round 2

Reviewer 2 Report

I ask the author to provide the information or difinition of regional level (R0-12) with relevant reference.

Other points were all resolved.

This manuscript is a resubmission of an earlier submission. The following is a list of the peer review reports and author responses from that submission.

Round 1

Reviewer 1 Report

General comments:

The topic of this manuscript is of high interest to understand the diagnostic performance of routine CT and CT enterography in estimating surgical and pathological PCI. However, the method section should be improved and expended to provide more detail information of the study design. Moreover, providing the comparison of diagnostic performance of rCT and CTE in the four intestinal regions and all 13 abdominopelvic regions is recommended to better represented the study aim.

Specific comments:

[Abstract]

Line 51-52: Please move the abbreviation “MDCT” into the right place.

Line54: Please briefly introduce CT enteroclysis to the readers.

Line 58: Please briefly introduce the differences and similarities of CT enteroclysis and CT enterography.

Line 61-64:The authors compared the diagnostic performance of rCT, CTE in predicting surgical and pathological PCI, aiming to demonstrate that CTE improves small bowel visualization as compared to rCT (line 265-266). Indeed, as stated in the line 54-55, “CT enteroclysis as a reliable preoperative mapping of the extent and distribution of PC in the small bowel and mesentery”. Therefore, it would be of interests that the authors could also provide the diagnostic performance of rCT and CTE in regions 9-12 (the four intestinal regions), in addition to those in all 13 regions.

[Materials and Methods]

Line 67-68: It seems that this was a prospectively study with all the patients signed an informed consent prior to enrolment but the Figure 1 stated that this was a retrospective study. Please indicate the time of signing the informed consent and the period of doing the image interpretation and analysis in the text and in Figure 1.

Line 72: According to the authors, data was analyzed retrospectively. What does it mean? Does it mean that the imaging data and surgical PCI was scored without blinded to pathological findings?

Line 79-80: In the study, there were two groups of patients, “Routine CT” and ‘CT Enterography”. Please add how patients were put into the two different groups in the method section and Figure 1, and discuss the possible selection bias in the discussion section.

Line 102: Table 1: Please provide comparison of characteristics for the two groups of patients, “Routine CT” and ‘CT Enterography”.   

Line 118 Table 2: The CT images were acquired using the same CT scanner but with different imaging protocol and acquisition parameters between the two groups of patients. Please provide reasons for the study design. Additionally, the slice thickness for patients in the CT Enterography group was smaller than that for patients in the Routine CT group. Readers would assume that the diagnostic performance between rCT and CTE would mainly arise from the image resolution as the thinner the slice the better the resolution is. Please clarify this issue in the discussion section.

Please also correct the abbreviation error for computed tomography in the first row first column of Table 1.

Line 156-157: According to the authors, the pathological PCI was assessed by two surgeons. However, it is not clear whether the average or the consensus score was used for the final statistics.

Line 160: Please briefly describe the scoring protocol for the pathological PCI in the reference [9], or cited another English article.

Line 167: 2.7.1 Concordance: Please provide reference(s) for the Lin’s correlation coefficient of agreement. Also, the criteria to interpretate the levels of agreement for the Lin’s correlation coefficient.

[Result]

Line 189: Please provide reference for interpretating “medium-magnitude” concordance for the Lin coefficient in the statistically section.

Line 275: Figure 2: Please make it clear whether the two CT images were obtained from the same or different patient. By the way, please provide arrows or arrowheads to indicate the peritoneal tumors.

Line 229-232: Please move the criteria for diagnostic index into the method section and provide the reference(s) for the interpretation.

[Discussion]

Line 271-273: Please provide the diagnostic performance of four intestinal regions (regions 9-12) vs. all abdominopelvic regions (region 0-12 or region 0-8) to support that appropriate intestinal distension gave better performance for estimating pathological PCI.  

Line 325-327: This study retrospectively analyzed patients underwent CRS between March 2011 and July 2017 and the CT images were interpreted retrospectively by the same radiologist, not experienced in patients with peritoneal carcinomatosis. Please provide the period of imaging interpretation in the method section and indicate whether the radiologist was blinded to the surgical or pathological PCI or not.  

Limitation section: Please also include the limitations below:

  • Patients were of the same tumor origin but may have different tumor extend between the two experiment groups.
  • Possible errors arising from the different imaging protocols for rCT and CTE
  • Selection bias for assigning patients into the two different groups

Author Response

Dr. Emmanuel Andrès 

Editor-in-Chief

Journal of Clinical Medicine

27 September 2021

“CT Enterography for Preoperative Evaluation of Peritoneal Carcinomatosis Index in Advanced Ovarian Cancer”

Ref: jcm-1387095

Dear Dr. Emmanuel Andrès,

I have received your mail with the reviewer´s reports about our work. Attached please find the reviewers’ lined comments along with my comments explaining why I agree or disagree with their opinions.

Response to Reviewer 1 Comments

Line 51-52: Please move the abbreviation “MDCT” into the right place.

Line 54: Please briefly introduce CT enteroclysis to the readers.

Line 58: Please briefly introduce the differences and similarities of CT enteroclysis and CT enterography.

Response:

Thank you for your valuable observations. I have introduced this paragraph in order to clarify your initial statements.

Multidetector Computed Tomography (MDCT) is the current imaging modality of choice for peritoneal carcinomatosis evaluation. Routine CT (rCT) is the usual technique to assess the extension studies of abdominal neoplasms, its main limitation is the inability to assess mesenteric involvement and small peritoneal deposits in the intestinal serosa [4]. A previous study identified CT enteroclysis (enteral contrast is introduced via a nasojejunal tube placed fluoroscopically prior to CT examination) as a reliable preoperative mapping of the extent and distribution of PC in the small bowel and mesentery [5]. According to another report, using CT enterography (where patients drink oral contrast) as a technical modification would improve the limitations of routine CT [6]. Enteral contrast is employed in both techniques to dilate the intestinal loops. However, enteroclysis requires the placement of a nasojejunal tube under fluoroscopic control, which increases radiation exposure and demands more ward and radiological time.

Line 61-64: The authors compared the diagnostic performance of rCT, CTE in predicting surgical and pathological PCI, aiming to demonstrate that CTE improves small bowel visualization as compared to rCT (line 265-266). Indeed, as stated in the line 54-55, “CT enteroclysis as a reliable preoperative mapping of the extent and distribution of PC in the small bowel and mesentery”. Therefore, it would be of interests that the authors could also provide the diagnostic performance of rCT and CTE in regions 9-12 (the four intestinal regions), in addition to those in all 13 regions.

Response:

The purpose of this study is to compare the diagnostic performance of routine CT (rCT), CT Enterography (CTE) and surgical in PCI scoring with surgical and pathological reference standards, and to analyse the diagnostic accuracy for the detection of lesions at the regional level (R0 to R8 and R9 to R12) compared to reference standards, in patients with tubo-ovarian peritoneal carcinomatosis.

[Materials and Methods]

Line 67-68: It seems that this was a prospectively study with all the patients signed an informed consent prior to enrollment but the Figure 1 stated that this was a retrospective study. Please indicate the time of signing the informed consent and the period of doing the image interpretation and analysis in the text and in Figure 1.

Line 72: According to the authors, data was analyzed retrospectively. What does it mean? Does it mean that the imaging data and surgical PCI was scored without blinded to pathological findings?

Line 79-80: In the study, there were two groups of patients, “Routine CT” and ‘CT Enterography”. Please add how patients were put into the two different groups in the method section and Figure 1, and discuss the possible selection bias in the discussion section.

Response:

This prospective study enrolled 148 consecutive patients treated by cytoreductive surgery in our hospital (a teaching university hospital), who were previously referred to the radiology service for staging or preoperative assessment (routine CT or CT Enterography) of peritoneal carcinomatosis, between March 2011 and July 2017. The study protocol was approved by the Institutional Review Board of our hospital and informed consent was obtained from each subject prior to the rCT or CTE examination. Surgical candidates underwent surgery within one or two months of imaging. Tumor characteristics were obtained from the medical records after the pathological examination, including tumour stage, grade, and histologic subtype. Pathological proven tubo-ovarian (epithelial ovarian cancer) or peritoneal cancer (n=128) was an inclusion criteria. Six patients were excluded as recurrent ovarian cancer (n=3), the interval between CT and CRS was > 30 days (n=1), imaging studies were performed without intravenous contrast (n=1), and there was a lack of surgical or histological reports (n=1). After establishing the selection criteria, we had a sample of 122 consecutive patients. In the first two years of the study, routine CT was performed on all participants (n=42), but following the apparent limitations of this technique, CT enterography was used as an optimised technique from July 2013 (n=80).

Image interpretation

The clinical report and PCI score were performed by a single radiologist with experience in abdominal imaging, immediately after the test was performed. The surgical team and the pathologist had access to the clinical report but were blinded to the result of the PCI score. 

Line 102: Table 1: Please provide comparison of characteristics for the two groups of patients, “Routine CT” and ‘CT Enterography”. 

rCT (n=42)

CTE (n=80)

Age (years)

Median

60

65

Range

27 – 44

31 – 84

CA 125

Median

276

368

Range

27 – 890

30 - 1200

Tumor origin

Ovarian

40 (95,2%)

76 (95%)

Fallopian tube

2 (4,8%)

2 (2,5%)

Peritoneum

0 (0%)

2 (2,5%)

Histologic findings

Serous cystoadenocarcinoma

32 (76,2%)

64 (80%)

Mucinosus  cystoadenocarcinoma

4 (9,5%)

6 (7,5%)

Endometrioid adenocarcinoma

6 (14,3%)

10 (12,5%)

Histological grade

Unknown

2 (5%)

6 (7,5%)

G1

0 (0%)

4 (5%)

G2

8 (20%)

6 (7,5%)

G3

31 (75%)

64 (80%)

Adverse Effects on CT

None

-

106 (87%)

Nausea

-

9 (7%)

Diarrhea

-

4 (3%)

Intestinal subocclusion

-

4 (3%)

Radiological PCI

1-10

34 (81%)

44 (55%)

11-20

8 (19%)

28 (35%)

>20

0 (0%)

8 (10%)

Surgical PCI

1-10

20 (47,6%)

26 (32,5%)

11-20

14 (33,4%)

28 (35%)

>20

8 (19%)

26 (32,5%)

Pathological PCI

1-10

30 (71,5%)

38 (47,5%)

11-20

8 (19%)

28 (35%)

>20

4 (9,5%)

14 (17,5%)

Cytoreduction

CC-0

34 (81%)

76 (95%)

CC-1 (<2cm)

4 (9,4%)

4 (5%)

CC-2 (2,6-5 cm)

2 (4,8%)

0 (0%)

CC-3 (>5cm)

2 (4,8%)

0 (0%)

Table 1. Clinicopathologic, Radiological and Surgical Characteristics of the 122 Patients with Advanced Ovarian Cancer Treated by Cytoreductive Surgery.

Line 118: Table 2: The CT images were acquired using the same CT scanner but with different imaging protocol and acquisition parameters between the two groups of patients. Please provide reasons for the study design. Additionally, the slice thickness for patients in the CT Enterography group was smaller than that for patients in the Routine CT group. Readers would assume that the diagnostic performance between rCT and CTE would mainly arise from the image resolution as the thinner the slice the better the resolution is. Please clarify this issue in the discussion section.

Response:

  1. Discussion

CT Enterography is an optimised imaging technique that combines thinner slice imaging with higher image resolution and adequate distension of the intestinal loops. We use it to overcome the limitations of routine CT. Little or no distension of the intestinal loops makes differentiation from serous implants or mesenteric nodules impossible, and poor image resolution makes assessing perihepatic and pelvic peritoneal structures difficult. In this study, we try to demostrate that a properly optimised MDCT imaging protocol can be a very useful tool in the initial selection of patients, without the need for MRI or PET-CT.

In patients with CP, the risk of intestinal subocclusion is higher; however, we observed that CTE was a safe method in these patients. Despite the use of a large volume of oral contrast, only 3% of our patients experienced this problem, which was managed with conservative treatment.

Line 156-157: According to the authors, the pathological PCI was assessed by two surgeons. However, it is not clear whether the average or the consensus score was used for the final statistics.

Response:

This is determined at the time of the laparotomy. Surgical exploration and palpation are performed to precisely identify the size and distribution of the tumour deposits [8]. It is also checked whether optimal cytoreduction (OCR) was achieved or not. The PCI assessment was performed by the two experienced surgeons, and the final score was established by consensus.

Line 160: Please briefly describe the scoring protocol for the pathological PCI in the reference [9], or cited another English article.

Response 

The pathological anatomy lab quantifies PCI based on the same criteria of tumour size and location; the PCI assessment was performed by the same pathologist. The approach followed a similar pattern to that detailed in another paper [9]. In brief, the surgeon labelled each section of the peritoneum in the en bloc specimen and sent it to the pathologist as a separate specimen. The pathologists discussed the size of the main tumour deposit as well as the presence of other minor deposits in each area. In the case of very small deposits, surgeons marked or sent the suspicious areas separately. One or more sections were extracted from the biggest nodule, as well as one or more sections from confluent deposits. The existence of lymph nodal disease was checked in the sub-peritoneal fat. Likewise, the omenta were inspected for the existence of lymph nodes. In the absence of gross disease, sections were taken every 2 to 5 cm, but this was not binding.

Line 167: 2.7.1 Concordance: Please provide reference(s) for the Lin’s correlation coefficient of agreement. Also, the criteria to interpretate the levels of agreement for the Lin’s correlation coefficient.

Response

Lin's correlation coefficient of agreement (rho_c) measures the agreement on a continuous measurement obtained by two people or methods [10, Steichen2002]. In this study, it was used to analyze the agreement of the total estimated magnitude of the PCI obtained by each one of the four measurement methods. The concordance correlation coefficient combines precision and accuracy measurements to determine the extent to which observed data deviate from the perfect concordance. The value of the Lin coefficient increases with proximity to the perfect accuracy and precision of the data. CCC quantifies the agreement ranging from −1 to 1, with perfect agreement at 1. CCC has the following classification according to strength of agreement (theoretical): >0.99 almost perfect, 0.95–0.99 substantial, 0.90–0.95 moderate, and <0.90 poor  [McBride2005].

[Result]

Line 189: Please provide reference for interpretating “medium-magnitude” concordance for the Lin coefficient in the statistically section.

Response

Lin’s concordance correlation was < 0,90 for all comparisons. Using the cutoffs proposed by McBride (McBride2005) this figure indicates poor agreement. The degrees of agreement (rCT and CTE) were lower when using the surgical PCI as the standard reference. The CTE, demonstrated a greater degree of concordance with pathological PCI (rho_c = 0.86) than did surgical and rCT (rho_c = 0,78 y 0,62), although it was still less than the <0.90 cutoff used to define poor agreement.

Line 275: Figure 2: Please make it clear whether the two CT images were obtained from the same or different patient. By the way, please provide arrows or arrowheads to indicate the peritoneal tumors.

Response

(a)

(b)

Figure 2. Axial slice images from different patients are shown to compare the degree of intestinal distention obtained by: (a) Routine CT; (b) CT Enterography section at the mid abdomen of a patients with peritoneal carcinomatosis show the difference between the non-optimized and optimized imaging techniques, with the CTE an adequate distention of the intestinal loops is achieved.  (rCT: Routine CT, CTE: CT Enterography), omental cake (arrowhead).

Line 229-232: Please move the criteria for diagnostic index into the method section and provide the reference(s) for the interpretation.

Response:

One way of interpreting the area under the ROC curve is that a test with an area greater than 0.9 has high accuracy, while 0.7–0.9 indicates moderate accuracy, 0.5–0.7, low accuracy and 0.5 a chance result (Fischer2003).

[Discussion]

Line 271-273: Please provide the diagnostic performance of four intestinal regions (regions 9-12) vs. all abdominopelvic regions (region 0-12 or region 0-8) to support that appropriate intestinal distension gave better performance for estimating pathological PCI. 

Response

Sensitivity

(95% CI)

Specificity

(95% CI)

Curved area ROC

(95% CI)

Surg

Path

Surg

Path

Surg

Path

R0 a R12

rCT

32%

(24 - 40)

39%

(29 - 48)

97%

(94 - 100)

96%

(93 - 99)

65%

(59 - 70)

67%

(61 - 74)

CTE

64%

(58 - 70)

71%

(65 - 77)

89%

(85 - 93)

82%

(77 -86)

76%

(71 - 81)

76%

(71 - 82)

R0 a R8

rCT

40%

(31 - 49)

44%

(34 - 55)

95%

(90 - 100)

93%

(88 - 98)

68%

(61 - 75)

69%

(61 - 76)

CTE

73%

(67- 79)

77%

(70 - 83)

84%

(78 - 90)

78%

(72 -84)

78%

(72 - 84)

77%

(71 - 83)

R9 a R12

rCT

NC*

NC*

NC*

NC*

NC*

NC*

CTE

39%

(28 - 51)

44%

(28 - 61)

97%

(93 - 100)

88%

(82 - 94)

68%

(60 - 75)

66%

(55 - 77)

Diagnostic performance in detecting regional involvement was similar when compared to surgery and pathological anatomy, however CTE performed better than rCT. CTE had a moderate diagnosis accuracy for regions 0 to 8 and the global assessment (regions 0 to 12). CTE had low accuracy in intestinal areas, and rCT could not be estimated.

Line 325-327: This study retrospectively analyzed patients underwent CRS between March 2011 and July 2017 and the CT images were interpreted retrospectively by the same radiologist, not experienced in patients with peritoneal carcinomatosis. Please provide the period of imaging interpretation in the method section and indicate whether the radiologist was blinded to the surgical or pathological PCI or not. 

Response

The research was limited by the fact that the images were read by a single radiologist with limited experience in peritoneal carcinomatosis, patients were of the same tumour origin but may have had different tumour extents between the two experiment groups, and possible errors arising from the different imaging protocols for rCT and CTE selection bias for assigning patients into the two different groups. 

I believe that I have made the most appropriate and effective corrections so that the readers of the journal will find this work interesting and useful for daily clinical practice. I hope you will consider our work again so that it can be published in your accredited journal.

We look forward to hearing from you at your earliest convenience.

Yours sincerely,

Katty Delgado MD PhD

Reviewer 2 Report

Thank you for your work. It was a pleasure to read it. I have some comments, and hope that you will be able to reply.

1. In the introduction - you mentioned "According to another report, 55 using CT enterography as a technical modification would improve the limitations of rou- 56 tine CT [6]."

This article talks about comparison between CT and MRI, there is no mention to CTE. please advise.

2. CTE is not used as common practice in most of the centers. why did you chose to use it, and how often you do that? 

3. In figure 1 - the last line of the exclusion criteria can be partially seen.

4. Table 1 is very disorganized. No demographic characteristics other than age. The imaging technique is not related to patients characteristics, nor the PCI and the completion of cytoreduction - those belong to surgical characteristics, and the imaging technique should be added to table 2.

5. There is no need to include section 2.2. The readers of this type of article know very well the interpretation of PCI.

6. The small bowel part of the PCI is composed only by 4 out of 13 areas. According to your article, we can better visualize the small bowel with CTE, yet to compare the total PCI without pointing to the small bowel. please explain

7. A very important point in the discussion - the radiologist who red the CT scan was with limited experience in peritoneal carcinomatosis. This is a major key point in your limitation and should have been avoided.

Again, thank you for your work. I will appreciate if you will be able to reply to my points.

Author Response

Dr. Emmanuel Andrès 

Editor-in-Chief

Journal of Clinical Medicine

27 September 2021

“CT Enterography for Preoperative Evaluation of Peritoneal Carcinomatosis Index in Advanced Ovarian Cancer”

Ref: jcm-1387095

Dear Dr. Emmanuel Andrès,

I have received your mail with the reviewer´s reports about our work. Attached please find the reviewers’ lined comments along with my comments explaining why I agree or disagree with their opinions.

Response to Reviewer 2 Comments

  1. In the introduction - you mentioned "According to another report, 55 using CT enterography as a technical modification would improve the limitations of routine CT [6].” This article talks about comparison between CT and MRI, there is no mention to CTE. please advise.

Response 1:

Thank yhou for your observation but as you can read in final section of the discussion:

Torkzad et al. [6] : “Our study suffers some limitations. One as mentioned before is that only two radiologists have read the images. Another problem is the techniques used. We have used only one series of sequences for MRI and one for CT. Modifications of both techniques such as CT enterography or addition of high resolution pelvic MRI might have changed results for some sites. “

  1. CTE is not used as common practice in most of the centers. why did you chose to use it, and how often you do that?

Response 2:

We picked this imaging technique because we believe it is the best for achieving appropriate distension of the intestinal loops, which is critical for determining the extent of peritoneal carcinomatosis. It's not a common technique, but it's quite effective and simple to apply, and we use it to examine non-acute intestinal haemorrhages and any cases of peritoneal carcinomatosis in which we want to quantify the tumour burden.

  1. In figure 1 - the last line of the exclusion criteria can be partially seen.

Response 3: 

Consecutive sampling of patients undergoing cytoreductive surgery between 3/2011 and 7/2017 (N=148)

Inclusion criteria

-        rCT or CTE studies performed at our radiology department

-        Pathological diagnosis of tubo-ovarian (epithelial ovarian cancer) or peritoneal cancer

-          

Potentially eligible patients   N = 128

Exclusion criteria

-        Recurrent ovarian cancer (n=3)

-        Interval between CT and CRS was >30 days (n = 1)

-        Imaging studies performed without intravenous contrast (n=1)

-        No surgical or histological reports (n=1)

Eligible patients          

N = 122

March 2011 – June 2013:

Routine CT (n = 42)

July 2013 – July 2017

CT Enterography (n = 80)

  1. Table 1 is very disorganized. No demographic characteristics other than age. The imaging technique is not related to patients characteristics, nor the PCI and the completion of cytoreduction - those belong to surgical characteristics, and the imaging technique should be added to table 2.

Response 4:

You are right, we have modified table 2 according to your suggestions

rCT (n=42)

CTE (n=80)

Age (years)

Median

60

65

Range

27 – 44

31 – 84

CA 125

Median

276

368

Range

27 – 890

30 - 1200

Tumor origin

Ovarian

40 (95,2%)

76 (95%)

Fallopian tube

2 (4,8%)

2 (2,5%)

Peritoneum

0 (0%)

2 (2,5%)

Histologic findings

Serous cystoadenocarcinoma

32 (76,2%)

64 (80%)

Mucinosus  cystoadenocarcinoma

4 (9,5%)

6 (7,5%)

Endometrioid adenocarcinoma

6 (14,3%)

10 (12,5%)

Histological grade

Unknown

2 (5%)

6 (7,5%)

G1

0 (0%)

4 (5%)

G2

8 (20%)

6 (7,5%)

G3

31 (75%)

64 (80%)

Adverse Effects on CT

None

-

106 (87%)

Nausea

-

9 (7%)

Diarrhea

-

4 (3%)

Intestinal subocclusion

-

4 (3%)

Radiological PCI

1-10

34 (81%)

44 (55%)

11-20

8 (19%)

28 (35%)

>20

0 (0%)

8 (10%)

Surgical PCI

1-10

20 (47,6%)

26 (32,5%)

11-20

14 (33,4%)

28 (35%)

>20

8 (19%)

26 (32,5%)

Pathological PCI

1-10

30 (71,5%)

38 (47,5%)

11-20

8 (19%)

28 (35%)

>20

4 (9,5%)

14 (17,5%)

Cytoreduction

CC-0

34 (81%)

76 (95%)

CC-1 (<2cm)

4 (9,4%)

4 (5%)

CC-2 (2,6-5 cm)

2 (4,8%)

0 (0%)

CC-3 (>5cm)

2 (4,8%)

0 (0%)

Table 1. Clinicopathologic, Radiological and Surgical Characteristics of the 122 Patients with Advanced Ovarian Cancer Treated by Cytoreductive Surgery.

  1. There is no need to include section 2.2. The readers of this type of article know very well the interpretation of PCI.

Response 5: Totally agree with your comment, we have eliminated this section.

  1. The small bowel part of the PCI is composed only by 4 out of 13 areas. According to your article, we can better visualize the small bowel with CTE, yet to compare the total PCI without pointing to the small bowel. please explain

Response 6:

Sensitivity

(95% CI)

Specificity

(95% CI)

Curved area ROC

(95% CI)

Surg

Path

Surg

Path

Surg

Path

R0 a R12

rCT

32%

(24 - 40)

39%

(29 - 48)

97%

(94 - 100)

96%

(93 - 99)

65%

(59 - 70)

67%

(61 - 74)

CTE

64%

(58 - 70)

71%

(65 - 77)

89%

(85 - 93)

82%

(77 -86)

76%

(71 - 81)

76%

(71 - 82)

R0 a R8

rCT

40%

(31 - 49)

44%

(34 - 55)

95%

(90 - 100)

93%

(88 - 98)

68%

(61 - 75)

69%

(61 - 76)

CTE

73%

(67- 79)

77%

(70 - 83)

84%

(78 - 90)

78%

(72 -84)

78%

(72 - 84)

77%

(71 - 83)

R9 a R12

rCT

NC*

NC*

NC*

NC*

NC*

NC*

CTE

39%

(28 - 51)

44%

(28 - 61)

97%

(93 - 100)

88%

(82 - 94)

68%

(60 - 75)

66%

(55 - 77)

Diagnostic performance in detecting regional involvement was similar when compared to surgery and pathological anatomy, however CTE performed better than rCT. CTE had a moderate diagnosis accuracy for regions 0 to 8 and the global assessment (regions 0 to 12). CTE had low accuracy in intestinal areas, and rCT could not be estimated.

  1. A very important point in the discussion - the radiologist who read the CT scan was with limited experience in peritoneal carcinomatosis. This is a major key point in your limitation and should have been avoided.

Response 7:

Yes, we would have preferred to avoid this, but not all centres have radiologists with experience in peritoneal carcinomatosis and obtaining a second external reader makes it difficult to complete a study of this type, which we consider to be one of the study's major limitations; however, we will look for ways to overcome this limitation in future research.

I believe that I have made the most appropriate and effective corrections so that the readers of the journal will find this work interesting and useful for daily clinical practice. I hope you will consider our work again so that it can be published in your accredited journal.

We look forward to hearing from you at your earliest convenience.

Yours sincerely,

Katty Delgado MD PhD

Round 2

Reviewer 1 Report

General comments:

Overall, the authors addressed the issues I listed and the revision was nicely done. However, I urge the authors to advice a native English-speaking editor to improve the manuscript. Also, they did not address the issue of their study design between rCT and CTE groups which are critical for readers to better understand the true value of CTE.

Specific comments:

  1. The better performance of CTE over rCT may arise from the study design but not entirely from enteral contrast administration itself (Figure 1 and Table 2). Indeed, confounding factors were not well controlled in the study (please see the comparison table below). Please discuss how the differences between groups might affect the outcome or add references to support the assumptions that slice thickness and CT scan as well as the administration of anti-peristaltic agent would not affect the diagnostic performance of PCI scoring. Alternatively, add these to the limitation section.

rCT

CTE

Study period

Year 2001-2013

Year 2003-2017

Radiologist experience

Limited

At least 2 years of experiences

CT protocol

5mm thickness

Only portal venous phase

3mm thickness

Biphasic

Bowel distension

800ml water

1800ml mannitol solution

Anti-peristaltic agent

-

+

  1. In Figure 2, the authors aimed to show the difference between the non-optimized and optimized imaging techniques, with the CTE an adequate distention of the intestinal loops is achieved. However, the revised Figure 2b shows bulky omental deposits but not tumor deposits on the intestinal wall. Theoretically, bulky omental deposits could be clearly demonstrated even in rCT. I would suggest the authors to revise it again.
  2. Line 51-52: The abbreviation “MDCT” did not move to the correct place in the text.
  3. Table 4: Please correct the error for the PCI score for Moderate PCI to 11-20. Also, use “Mod PCI for the 1st column title for “Moderate PCI”.
  4. Table 5: Please change the PCI score for High PCI to > 20. Also, readers may not be familiar with the word “curved area ROC”. Area under the ROC curve or AUC may be more commonly used.

Author Response

Response to Reviewer 1 Comments

General comments:

Overall, the authors addressed the issues I listed and the revision was nicely done. However, I urge the authors to advice a native English-speaking editor to improve the manuscript. Also, they did not address the issue of their study design between rCT and CTE groups which are critical for readers to better understand the true value of CTE.

We, the authors of the article, commit ourselves to make use of MDPI Author Services - English Editing Services after all revisions and changes have been made, so please understand and excuse the inconvenience.

Point 1: The better performance of CTE over rCT may arise from the study design but not entirely from enteral contrast administration itself (Figure 1 and Table 2). Indeed, confounding factors were not well controlled in the study (please see the comparison table below). Please discuss how the differences between groups might affect the outcome or add references to support the assumptions that slice thickness and CT scan as well as the administration of anti-peristaltic agent would not affect the diagnostic performance of PCI scoring. Alternatively, add these to the limitation section.

Response 1:

In our study, these two imaging techniques were performed in 122 consecutive patients, showing that CT enterography is more accurate than routine CT in the detection of peritoneal implants in all thirteen regions. In particular, the sensitivity of CTE was significantly higher than that of rCT for the detection of lesions in regions 0 to 8. The specificity of CTE was significantly higher than that of rCT for the detection of peritoneal implants in regions 9 to 12.

Bowel distension:

The major limitation of routine abdominopelvic CT was the inadequate distension of the bowel loops; serous implants and mesenteric nodules were missed or confused with collapsed loops, resulting in the failure to detect lesions in regions 9 to 12. In order to improve this limitation, we used CTE as an optimised technique. CT enterography differs from routine abdominopelvic CT in that it makes use of thin sections and large volumes of enteric contrast material to better display the small bowel lumen and wall (Paulsen, 2006 RG). Although CT enterography is a powerful tool in the evaluation of small bowel disease, our study demonstrates a low diagnostic accuracy in the identification of implants in regions R9 to R12. 

CT protocol:

In relation to the CT technique, Fletcher (Fletcher2008AbdImg) explains that while initial CT enterography reports used a slice thickness of 5 mm, slice thicknesses between 1 and 3 mm are generally preferred with MDCT as they permit improved contrast and spatial resolution with acceptable tradeoffs and increased image noise. In our study, CTE has proved better diagnostic accuracy than rCT in the detection of implants in the regions (R0 to R8), suggesting that a thinner slice protocol allows greater definition of the porta hepatis and lesser omentum.

Our study had several limitations.

Radiologist experience:

First, our study design introduced some confounding factors; the images were read by a single radiologist. Previous studies [6,16] demonstrated the impact of experience in image interpretation in patients with peritoneal carcinomatosis; the radiologist's experience was zero in rCT cases, as opposed to two years of experience in CTE cases.

Anti-peristaltic agent:

Only in the CTE group were antiperistaltic agents used to eliminate peristalsis and reduce motion artefacts. Another limitation was that the patients had the same tumour origin but could have different tumour extensions between the two groups in the experiment.

Point 2: In Figure 2, the authors aimed to show the difference between the non-optimized and optimized imaging techniques, with the CTE an adequate distention of the intestinal loops is achieved. However, the revised Figure 2b shows bulky omental deposits but not tumor deposits on the intestinal wall. Theoretically, bulky omental deposits could be clearly demonstrated even in rCT. I would suggest the authors to revise it again.

Response 2:

(a)

(b)

(c)

(d)

Figure 2. Axial slice images from different patients are shown to compare the degree of intestinal distention obtained by: (a) and (b) Routine CT; (c) and (d) CT Enterography section at the pelvis and mid abdomen of a patients with peritoneal carcinomatosis show the difference between the non-optimized and optimized imaging techniques, with the CTE an adequate distention of the intestinal loops is achieved.  (rCT: Routine CT, CTE: CT Enterography), soft tissue nodules (arrowhead).

Point 3: Line 51-52: The abbreviation “MDCT” did not move to the correct place in the text.

Response 3:

Multidetector Computed Tomography (MDCT) is the current imaging modality of choice for peritoneal carcinomatosis evaluation. Routine CT (rCT) is the usual technique to assess the extension studies of abdominal neoplasms, its main limitation is the inability to assess mesenteric involvement and small peritoneal deposits in the intestinal serosa [4].

Point 4: Table 4: Please correct the error for the PCI score for Moderate PCI to 11-20. Also, use “Mod PCI for the 1st column title for “Moderate PCI”.

Low PCI

Mod PCI

High PCI

Surg

Path

Surg

Path

Surg

Path

Low PCI

1 - 10

rCT

(»)100 %

(») 93 %

(ß) 86 %

(ß) 75 %

(ß) 25 %

-

20

28

12

6

2

0

CTE

(») 92 %

(») 95 %

(ß) 64 %

(ß) 29 %

(ß) 8 %

-

24

36

18

8

2

0

Surgical

(») 65 %

-

(ß) 6 %

-

-

44

0

2

0

Mod PCI

11 - 20

rCT

-

(Ý) 7 %

(») 14 %

(») 25 %

(ß) 75 %

(ß) 100 %

0

2

2

2

6

4

CTE

(Ý) 3 %

(Ý) 5 %

(») 29 %

(») 57 %

(ß) 69 %

(ß) 71 %

2

2

8

16

18

10

Surgical

(Ý) 32 %

-

(») 50 %

(ß) 11 %

22

0

0

2

High PCI

>20

rCT

-

-

-

-

-

-

0

0

0

0

0

0

CTE

-

-

(Ý) 7 %

(Ý) 14 %

(») 23 %

(») 29 %

0

0

2

4

6

4

Surgical

(Ý) 3 %

(Ý) 44 %

(») 89 %

2

16

16

Point 5: Table 5: Please change the PCI score for High PCI to > 20. Also, readers may not be familiar with the word “curved area ROC”. Area under the ROC curve or AUC may be more commonly used.

Sensitivity

(95% CI)

Specificity

(95% CI)

AUC

(95% CI)

Surg

Path

Surg

Path

Surg

Path

Low PCI

1–10

rCT

100%

(69 - 100)

100%

(78 - 100)

27%

(6 - 61)

50%

(12 - 88)

64%

(50 - 77)

75%

(53 - 97)

CTE

92%

(64 - 100)

94%

(73 - 100)

63%

(42 - 81)

77%

(55 -92)

78%

(66 - 90)

86%

(75 - 96)

Surg

-

68%

(48 - 82)

-

96%

(82 - 100)

-

82%

(73 - 90)

Mod PCI

11-20

rCT

25%

(63 - 81)

25%

(63 - 81)

65%

(38 - 86)

82%

(57 - 96)

45%

(18 - 72)

54%

(28 - 80)

CTE

29%

(8- 58)

57%

(29 - 82)

62%

(41 - 80)

77%

(56 -91)

45%

(30 - 61)

67%

(51 - 83)

Surg

-

50%

(26 - 74)

-

72%

(56 - 85)

-

61%

(47 - 75)

High PCI

>20

rCT

NC*

NC*

NC*

NC*

NC*

NC*

CTE

75%

(19 - 99)

50%

(7 - 93)

72%

(55 - 86)

76%

(81 - 95)

74%

(48 - 99)

68%

(39 - 97)

Surg

-

89%

(52 - 100)

-

83%

(70 - 92)

-

86%

(74 - 98)

Round 3

Reviewer 1 Report

  1. The authors mentioned that they analyzed the diagnostic performance for PCI in routine CT (rCT), CT Enterography (CTE) at the regional level (R0 to R8 and R9 to R12) (see line 67-71). And in discussion section (see line 343-346), they summarized the difference of diagnostic performance using two methods. However, there was not relevant table or text in the result section which compared the regional difference of rCT and CTE.

The authors have the table below only in the rebutting letter but not in the revised manuscript. The table should be added into the manuscript if the authors preferred to keep the current text.

Sensitivity

(95% CI)

Specificity

(95% CI)

AUC

(95% CI)

Surg

Path

Surg

Path

Surg

Path

R0 to R12

rCT

32%

(24 - 40)

39%

(29 - 48)

97%

(94 - 100)

96%

(93 - 99)

65%

(59 - 70)

67%

(61 - 74)

CTE

64%

(58 - 70)

71%

(65 - 77)

89%

(85 - 93)

82%

(77 -86)

76%

(71 - 81)

76%

(71 - 82)

R0 to R8

rCT

40%

(31 - 49)

44%

(34 - 55)

95%

(90 - 100)

93%

(88 - 98)

68%

(61 - 75)

69%

(61 - 76)

CTE

73%

(67- 79)

77%

(70 - 83)

84%

(78 - 90)

78%

(72 -84)

78%

(72 - 84)

77%

(71 - 83)

R9 to R12

rCT

NC*

NC*

NC*

NC*

NC*

NC*

CTE

39%

(28 - 51)

44%

(28 - 61)

97%

(93 - 100)

88%

(82 - 94)

68%

(60 - 75)

66%

(55 - 77)

  1. In line 355-357, the authors wrote:

“In relation to the CT technique, Fletcher (Fletcher2008AbdImg) explains that while initial CT enterography reports used a slice thickness of 5 mm, slice thicknesses between 1 and 3 mm are generally preferred with MDCT as they permit improved contrast and spatial resolution with acceptable tradeoffs and increased image noise.’

In the sentence, the authors implied that rCT with slice thickness of 3mm yielded better image quality that those with slice thickness of 5mm. As such, the authors should not compare rCT of 5mm thickness with CTE of 3mm thickness. I strongly recommend the authors to add this limitation into the limitation section.

  1. It is not clear why the better diagnostic accuracy by CTE in Regions 0-8 suggested that a thinner slice protocol allows greater definition of the porta hepatis and lesser omentum in line 358-360. Please explain in details.

Author Response

Response to Reviewer 1 Comments - Round 3

  1. The authors mentioned that they analyzed the diagnostic performance for PCI in routine CT (rCT), CT Enterography (CTE) at the regional level (R0 to R8 and R9 to R12) (see line 67-71). And in discussion section (see line 343-346), they summarized the difference of diagnostic performance using two methods. However, there was not relevant table or text in the result section which compared the regional difference of rCT and CTE.

The authors have the table below only in the rebutting letter but not in the revised manuscript. The table should be added into the manuscript if the authors preferred to keep the current text.

I fully agree, it was a mistake in updating the document, I am very sorry.

3.1. Analysis of diagnostic performance based on lesion detection at the regional level

Considering the regional level analysis, 1586 abdominopelvic regions were tested (rCT: 546 and CTE: 1040). We found a total correlation (in terms of the presence or absence) between rCT and surgical examination in 356 out of 546 regions [64%, 242 out of 378 regions in R0 a R8 and 114 out of 168 regions in R9 a R12, for a total concordance in 86 out of 546 (16%) in a positive sense and in 270 out of 546 (49%) in a negative sense] and a correlation between rCT and pathologic score in 406 out of 546 regions [ 70%, 264 out of 378 regions in R0 a R8 and 142 out of 168 in R9 a R12, for a total concordance in 80 out of 546 (15%) in a positive sense and in 326 out of 546 (60%) in a negative sense].

Between CTE and surgical examination, we found a total correlation (in terms of the presence or absence) in 786 out of 1040 regions [77,5%, 558 out of 720 regions in R0 a R8 and 228 out of 320 regions in R9 a R12, for a total concordance in 354 out of 1040 (34%) in a positive sense and in 432 out of 1040 (42%) in a negative sense] and a correlation between rCT and pathologic score in 806 out of 1040 regions [ 77%, 556 out of 720 regions in R0 a R8 and 250 out of 320 in R9 a R12, for a total concordance in 296 out of 1040 (28%) in a positive sense and in 510 out of 1040 (49%) in a negative sense].

As for the diagnostic performance in detecting regional involvement was similar when compared to surgery and pathological anatomy, however CTE performed better than rCT. CTE had a moderate diagnosis accuracy for regions 0 to 8 and the global assessment (regions 0 to 12). CTE had low accuracy in intestinal areas, and rCT could not be estimated given the results obtained (true positives=0). (Table 3).

Table 3. Comparative Diagnostic Performance Between rCT and CTE vs surgical and pathological examination at regional level analyses

Sensitivity

(95% CI)

Specificity

(95% CI)

AUC

(95% CI)

Surg

Path

Surg

Path

Surg

Path

R0 a R12

rCT

32%

(24 - 40)

39%

(29 - 48)

97%

(94 - 100)

96%

(93 - 99)

65%

(59 - 70)

67%

(61 - 74)

CTE

64%

(58 - 70)

71%

(65 - 77)

89%

(85 - 93)

82%

(77 -86)

76%

(71 - 81)

76%

(71 - 82)

R0 a R8

rCT

40%

(31 - 49)

44%

(34 - 55)

95%

90 - 100)

93%

(88 - 98)

68%

(61 - 75)

69%

(61 - 76)

CTE

73%

(67- 79)

77%

(70 - 83)

84%

(78 - 90)

78%

(72 -84)

78%

(72 - 84)

77%

(71 - 83)

R9 a R12

rCT

NC*

NC*

NC*

NC*

NC*

NC*

CTE

39%

(28 - 51)

44%

(28 - 61)

97%

(93 - 100)

88%

(82 - 94)

68%

(60 - 75)

66%

(55 - 77)

  1. In line 355-357, the authors wrote:

“In relation to the CT technique, Fletcher (Fletcher2008AbdImg) explains that while initial CT enterography reports used a slice thickness of 5 mm, slice thicknesses between 1 and 3 mm are generally preferred with MDCT as they permit improved contrast and spatial resolution with acceptable tradeoffs and increased image noise.’

In the sentence, the authors implied that rCT with slice thickness of 3mm yielded better image quality that those with slice thickness of 5mm. As such, the authors should not compare rCT of 5mm thickness with CTE of 3mm thickness. I strongly recommend the authors to add this limitation into the limitation section. I agree with the correction, I have added the suggestion in the section on the limitations of the study.

Our study had several limitations. First, our study design introduced some confounding factors; the images were read by a single radiologist. Previous studies [6,23]   demonstrated the impact of experience in image interpretation in patients with peritoneal carcinomatosis; the radiologist's experience was zero in rCT cases, as opposed to two and a half years of experience in CTE cases. The slice thickness was not the same in the two radiological tests. Only in the CTE group were antiperistaltic agents used to eliminate peristalsis and reduce motion artefacts. Another limitation was that the patients had the same tumour origin but could have different tumour extensions between the two groups in the experiment.

  1. It is not clear why the better diagnostic accuracy by CTE in Regions 0-8 suggested that a thinner slice protocol allows greater definition of the porta hepatis and lesser omentum in line 358-360. Please explain in details.

I agree with the correction, I do not have sufficient arguments to demonstrate the best diagnostic performance in region 1 or 2, it is a conclusion based on experience, but without scientific support.

In our study, CTE has proved better diagnostic accuracy than rCT in the detection of implants in the regions (R0 to R8).